# Transient Stability Analysis of Direct Drive Wind Turbine in DC-Link Voltage Control Timescale during Grid Fault

**Qi Hu** ⬤, **Yiyong Xiong, Chenruiyang Liu \*, Guangyu Wang and Yanhong Ma**

National Key Laboratory of Science and Technology on Vessel Integrated Power System,
School of Electrical Engineering, Naval University of Engineering, Wuhan 430033, China;
qihu0725@163.com (Q.H.); xiongyiyong1989@163.com (Y.X.); hust_wang000@163.com (G.W.);
yanhongma0421@163.com (Y.M.)
**\*** Correspondence: lychee0825@outlook.com

**Abstract:** Transient stability during grid fault is experienced differently in modern power systems, especially in wind-turbine-dominated power systems. In this paper, transient behavior and stability issues of a direct drive wind turbine during fault recovery in DC-link voltage control timescale are studied. First, the motion equation model that depicts the phase and amplitude dynamics of internal voltage driven by unbalanced active and reactive power is developed to physically depict transient characteristics of the direct drive wind turbine itself. Considering transient switch control induced by active power climbing, the two-stage model is employed. Based on the motion equation model, transient behavior during fault recovery in a single machine infinite bus system is studied, and the analysis is also divided into two stages: during and after active power climbing. During active power climbing, a novel approximate analytical expression is proposed to clearly reveal the frequency dynamics of the direct drive wind turbine, which is identified as approximate monotonicity at excitation of active power climbing. After active power climbing, large-signal oscillation behavior is concerned. A novel analysis idea combining time-frequency analysis based on Hilbert transform and high order modes is employed to investigate and reveal the nonlinear oscillation, which is characterized by time-varying oscillation frequency and amplitude attenuation ratio. It is found that the nonlinear oscillation and even stability are related closely to the final point during active power climbing. With a large active power climbing rate, the nonlinear oscillation may lose stability. Simulated results based on MATLAB® are also presented to verify the theoretical analysis.

**Keywords:** direct drive wind turbine; grid fault; nonlinear oscillation; transient stability; time-frequency analysis; transient switch

## 1. Introduction

With an increasing penetration of wind power integrated into modern power systems, the dynamic issue of part grid tends to be dominated by wind turbines instead of traditional synchronous generators. Wind turbines have different dynamic characteristics than synchronous generators, resulting in the system experiencing different dynamic issues. Among different types of wind turbines, direct drives that have superior grid-connected performance are increasingly installed. However, due to the reverse distribution of wind resources and load centers, a large scale of direct drive wind turbines is installed in a weak AC grid, bringing in strong interaction between the wind turbine and AC grid. The strong dynamic interaction significantly challenges the safe and stable operation of the system, necessitating the analysis.

Existing studies have paid much attention to the stability issue resulting from the connection of wind turbines [1]. In [2–8], a small-signal oscillation problem related to wind turbines integrated into a high impedance AC grid is investigated. Due to a wide band control of equipment, oscillation is characterized by a multi-time scale, and oscillation

frequency ranges from hundreds of Hz to several Hz. Previous works have carried out detailed analyses about this. However, these works address the small-signal stability issue with disturbance around the equilibrium point, and a linearized system is applicable for analysis. In practice, faults, including wind turbine system faults and grid faults, are common. Fault diagnosis and resilient control for a wind turbine system is a research hotspot that has attracted massive research in recent years [9,10]. Except for this, transient issue analysis during grid fault is also worthy of research. In [11], the rotor angle stability of the synchronous generator affected by the dynamical characteristics of a wind turbine is analyzed. The work is carried out from the viewpoint that synchronous generators are dominant equipment, and the dynamics of wind turbines are only influential factors. This is reasonable at a relatively low penetration of wind power. However, with the increasing penetration of wind power, the dynamic issue of the part grid is dominated by wind turbines instead of synchronous generators. Transient issue faces new challenges and begins to be paid attention to. Transient stability dominated by the control of renewable energy generating units are investigated in [12–17]. Analysis results show that similar transient instability that is common in traditional synchronous generators also exists in PLL-synchronized converters. The transient stability can be explored from the accelerating and decelerating areas method. These analyses are based on a simplified control structure and attempt to reveal transient instability mechanisms. Yet, practical control of wind turbines is complex, even on a single time scale [3,8–11]. In this paper, transient behavior during fault recovery in DC-link voltage control timescale is studied, with complex practical control considered.

A deep understanding of an equipment's characteristics is the precondition of dynamic issue analysis. In order to investigate the dynamic behavior of a system dominated by renewable energy, kinds of equipment models are proposed [18–21]. The impedance model is developed and widely used in small-signal oscillation analysis. External characteristics of equipment are investigated through impedance frequency spectrum with specific control structure packing treatment [22–24]. At the time of bringing convenience, it has some difficulty in mechanism explanation of the relationship between specific control loop and oscillation. Based on this consideration, the motion equation model from the idea of Newtonian mechanics is proposed to deeply study equipment's characteristics [20,21]. By establishing the relationship between unbalanced powers and dynamics of internal voltage, the form of the motion equation model is similar to the rotor motion of a synchronous generator, and equivalent inertia and damping can be obtained. Thus, oscillation with increasing amplitude can be physically explored from the viewpoint of insufficient damping. However, the two models are both applicable for small signal analysis. Under large-signal disturbance, a new model is needed to study equipment's transient characteristics. Based on the advantage of the motion equation model in studying the equipment's characteristics, it is necessary to be popularized for the condition of large-signal disturbance. In this paper, the transient motion equation model in the DC-Link voltage control time scale is developed with transient switch control considered.

Although large signal analysis is difficult due to the non-negligible influence of non-linearity, kinds of meaningful methods are proposed to address the issue [25–32]. The methods based on computational intelligence may be powerful for the analysis and control of a complex, large-scale system [25–27]. However, they may have difficulty explaining the stability mechanism and influence factor concerned by this paper. Time-frequency analysis based on the Hilbert transform is usually employed to analyze low-frequency and sub-synchronous nonlinear oscillation in traditional power systems [28,29]. Based on data from transient simulations, instantaneous attributes of oscillation behavior can be identified. In addition to numerical analysis, the inclusion of higher-order terms is usually used to evaluate accurate modal characteristics that linear analysis can not provide [30,31]. Based on the Normal Form theory, higher-order modal interactions resulting from the influence of nonlinearity can be revealed. By combining the two methods, nonlinear oscillation can

be deeply investigated [32]. This paper draws lessons from the two methods and carries out large-signal oscillation analysis during fault recovery.

The rest part of this paper is organized as follows. In Section 2, transient switch control of direct drive wind turbine is investigated. Then motion equation model during fault recovery is developed in Section 3. Based on the developed model, transient behavior analysis in a simple system is carried out in Section 4. Finally, conclusions are drawn in Section 5.

## 2. Transient Switch Control of Direct Drive Wind Turbine

When grid faults occur, the wind turbine usually undergoes complex transient switch control to support the grid or protect the wind turbine itself. Figure 1 shows the typical auxiliary control and circuit referred to [33,34]. Due to this concerning issue, the control in electromagnetic time scale receives special attention.

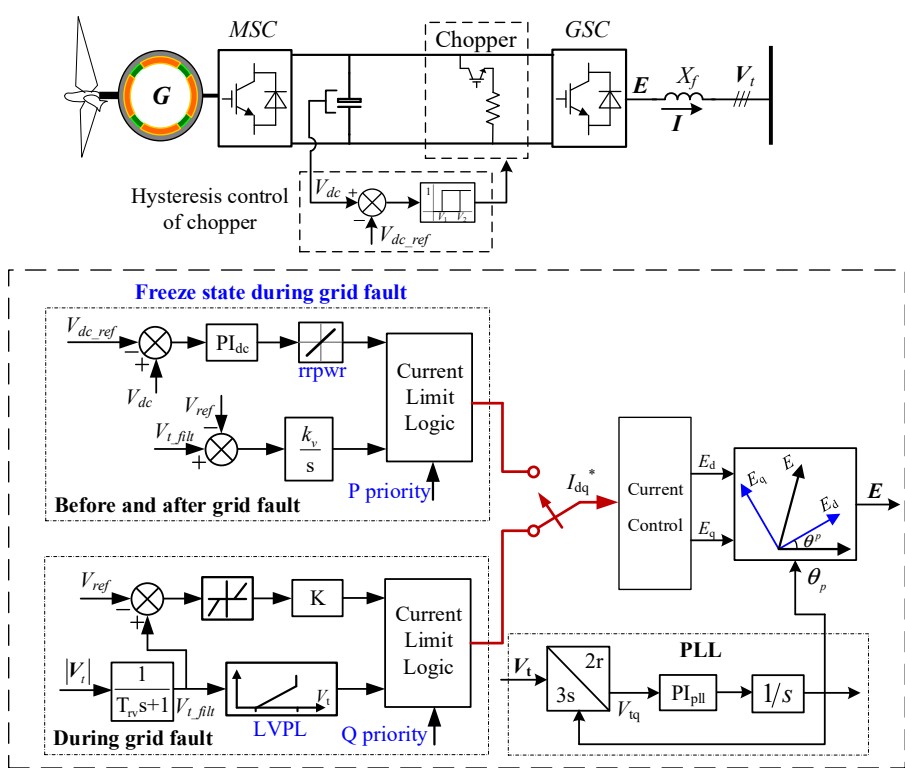

**Figure 1.** Typical auxiliary control and circuit in response to grid faults.

As shown in Figure 1, the whole process in response to grid faults can be divided into three stages according to that grid faults are detected and then cleared, which is as shown in Figure 2. Each stage employs a different control structure in order to satisfy different requirements.

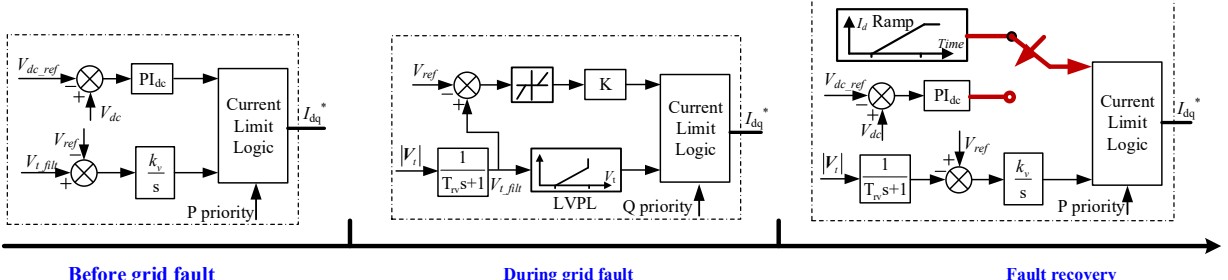

**Figure 2.** Transient switch control in response to grid fault.

Before grid fault, active and reactive current orders are controlled by DC-Link voltage control and terminal voltage control, respectively. Due to the limited capacity of the grid side converter, active power priority is usually utilized in current limit logic.

In the case that grid faults are detected, states of DC-Link voltage control and terminal voltage control before the grid fault are frozen first. Grid side converter takes the role of supporting grid voltage and injects reactive current as required by grid codes. In order to reduce the system stress during grid faults, the active current order is limited by a cap (upper limit) through Low Voltage Power Logic (LVPL) [34]. In normal operating conditions, there is no cap. When the voltage falls, a cap is calculated and applied. Thus, the dynamics of the active current order are influenced by the amplitude of terminal voltage during a deep grid fault. Referring to possible DC-Link overvoltage resulting from limited active power transfer, a hysteresis controller based on chopper-controlled resistors is employed to stabilize DC-Link voltage in the set narrowband.

When grid faults are cleared, active and reactive current orders are re-controlled by DC-Link voltage control and terminal voltage control, respectively. However, a ramp rate limit is applied to the active current order rate of increase to reduce system stress [34]. Since active current order during the grid fault is usually very small, it increases with time according to the ramp rate limit in a short time during fault recovery. When it reaches about the frozen value before the grid fault, the active current order begins to be adjusted by DC-Link voltage control. As a result, the transient process during fault recovery can be further divided into two stages: during and after active power climbing. A switched system should be employed to portray the transient behavior during fault recovery.

## 3. Developed Motion Equation Model

Since direct drive wind turbines employ a power electronic converter as a grid-connected interface, their transient characteristic is dominated by complex control. In order to physically study the transient characteristic, a motion equation model based on Newton mechanics is proposed, which establishes the relationship of internal voltage dynamics induced by unbalanced powers. Then the transient characteristic of the direct drive wind turbine can be explored from the equivalent motion driven by unbalanced powers. Concerning the transient switch control during fault recovery, the transient analysis should be divided into two stages: during and after active power climbing. The switched system should be employed to depict transient characteristics. During active power climbing, active current order increases with time according to the ramp rate limit, and DC-Link voltage control does not take effect. After active power climbing, active current order begins to be adjusted by DC-Link voltage control.

### 3.1. Motion Equation Model in Stage of Active Power Climbing

Based on Figure 1, the dynamics of the wind turbine's internal voltage in the stage of active power climbing are dominated by terminal voltage control and a phase-locked loop, as shown in Figure 3. Since the two control loops are in response to dynamics of terminal voltage and then adjust current orders, the modeling work is mainly composed of two parts. One is that the phase and amplitude dynamics of terminal voltage should be obtained through active and reactive power (*P*,*Q*). Based on this, a model can be developed in the form that dynamics are induced by unbalanced powers, and the model has good portability due to no relationship with the information of the network. The other is that internal voltage should be calculated through current orders since the internal voltage is selected to represent the external characteristic of the wind turbine.

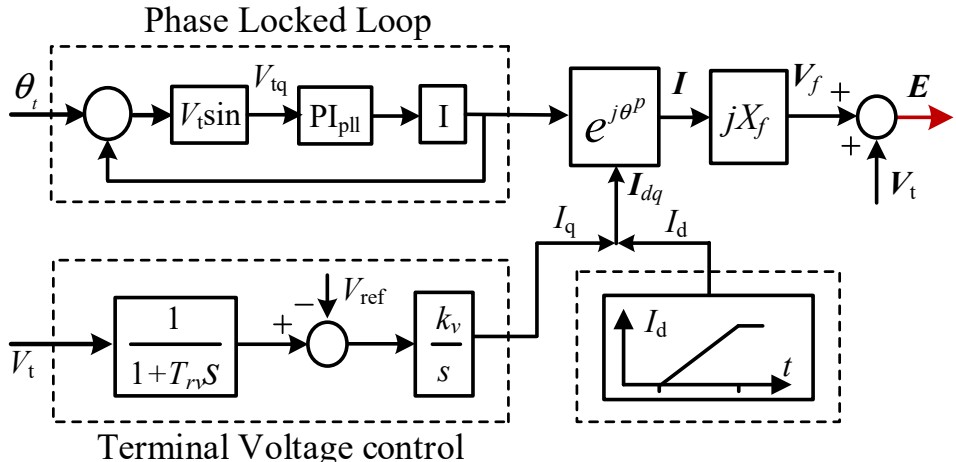

**Figure 3.** Dynamics of internal voltage in stage I.

First, the relationship of terminal voltage with active and reactive power output is calculated. According to the circuit topology in Figure 1, the output power is represented by

$$P = \frac{EV_t \sin(\theta_e - \theta_t)}{X_f} \tag{1}$$

$$Q = \frac{\left[E^2 - EV_t \cos(\theta_e - \theta_t)\right]}{X_f} \tag{2}$$

Then, combining (1) and (2), phase and amplitude dynamics of terminal voltage can be obtained by

$$\theta_t = \theta_e - \arctan\left[\frac{PX_f}{\left(E^2 - QX_f\right)}\right] \tag{3}$$

$$V_t = \frac{\sqrt{P^2 X_f{}^2 + \left(E^2 - QX_f\right)^2}}{E} \tag{4}$$

Thus, information on terminal voltage can be replaced by internal voltage and active and reactive power output.

Second, the internal voltage should be calculated through current orders. It is known that current orders adjusted by control loops are in the PLL reference frame. Based on the circuit relationship in Figure 1, the *dq* component of internal voltage can be calculated by

$$E_d = V_t \cos\theta_t^p - X_f I_q \tag{5}$$

$$E_q = V_t \sin\theta_t^p + X_f I_d \tag{6}$$

Through polar coordinates transformation, amplitude and phase (that is, relative to d-axis of PLL) of internal voltage can be obtained by

$$E = \sqrt{E_d{}^2 + E_q{}^2} \tag{7}$$

$$\theta_e^p = a\tan\left(\frac{E_q}{E_d}\right) \tag{8}$$

Due to the employed PLL synchronization, the phase of internal voltage is composed of two parts: the synchronous phase provided by PLL and the phase that is relative to PLL, as represented by

$$\theta_e = \theta_e^p + \theta_p \tag{9}$$

Based on the above deduction, the developed motion equation mode during active power climbing is shown in Figure 4. It is clearly seen that the dynamics of internal voltage can be studied from the equivalent motion driven by unbalanced active and reactive power.

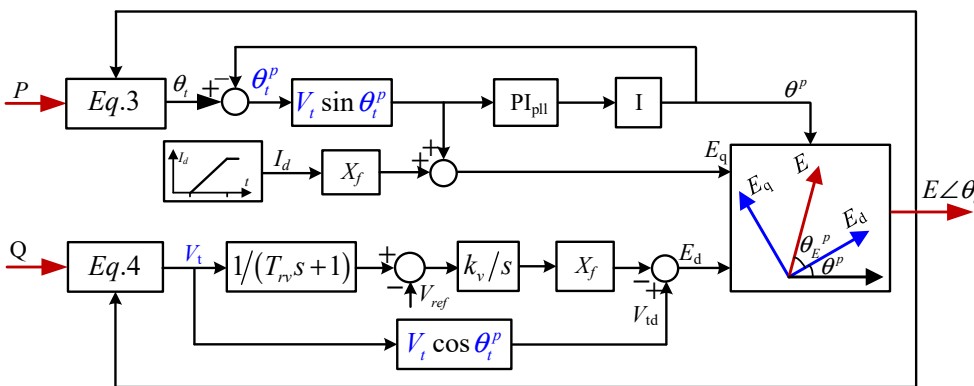

**Figure 4.** Motion equation mode in stage of active power climbing.

### 3.2. Motion Equation Model after Active Power Climbing

When the active current order approaches frozen value before grid fault, ramp rate limit will be out of action, and DC-Link voltage control begins to take effect. Thus the influence of DC-Link voltage control on dynamics of internal voltage should be considered after active power climbing.

In this case, when electromagnetic power injected into the power grid is not equal to feed power from the machine side, the DC-link capacitor will go through charging or discharging. Then the active current will be adjusted and thus significantly influence phase dynamics. This indicates that unbalanced active power drives the motion of phase, although the relationship between them is complex. Moreover, when DC-Link voltage exceeds the limit value, the chopper will take effect, and consumed power by the chopper should be taken into account. Based on Figure 4, the motion equation model after active power climbing can be easy to be obtained, as shown in Figure 5.

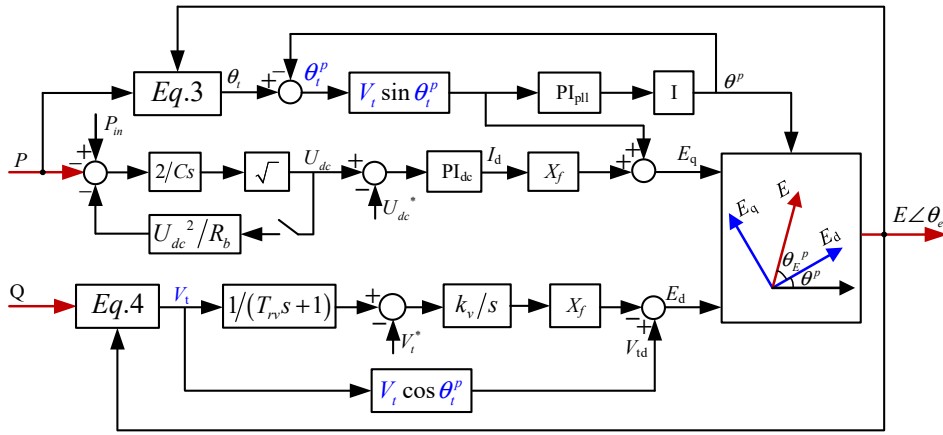

**Figure 5.** Motion equation model after active power climbing.

### 3.3. Equipment's Transient Characteristic Analysis

Based on the developed motion equation model in Figures 4 and 5, it is known that wind turbine is very different from traditional synchronous generators and its transient characteristic are much more complex, which can be concluded as

(1) Discontinuity. Unlike the synchronous generators that can employ a unified model for electromechanical transient analysis in different fault stages, the developed

motion equation model of the direct drive wind turbine is discontinuous due to transient switch control.

(2) Nonlinearity. In transient stability analysis of traditional power systems, nonlinearity mainly results from the network, which lies in the power angle curve, and the linear rotor motion model is used to depict the equipment's transient characteristic. However, the wind turbine's motion equation model is characterized by strong nonlinearity, and nonlinearity is mainly embodied in the following three aspects: polar transformation, PLL, and replacing terminal voltage information. The main types of nonlinearity are trigonometric and square functions.

(3) High order. Due to the complex control of wind turbines, the relationship between unbalanced power and internal voltage is characterized by high order. As a result, the inertia that is used to depict the relationship between unbalanced active power and phase dynamics is variable. This is different from a synchronous generator, which has constant inertia.

(4) Strong Coupling. In a wind turbine, phase dynamics are strongly coupled with amplitude dynamics, and the coupling that mainly results from the control of the wind turbine is implemented in an orthogonal coordinate system, while amplitude and phase are obtained from the polar coordinate system. Compared with a synchronous generator that directly controls amplitude and phase, the coupling in a wind turbine is stronger.

## 4. Transient Analysis in Single-Machine Infinite-Bus System

Based on the developed motion equation model, transient analysis during fault recovery in a typical single-machine infinite-bus (SMIB) system shown in Figure 6 is carried out. A three-phase ground fault is set at one line, and after a certain time, the faulted line is cut off. In this paper, we assume that a stable operating point has been achieved during a grid fault, and transient behavior during fault recovery is mainly concerned.

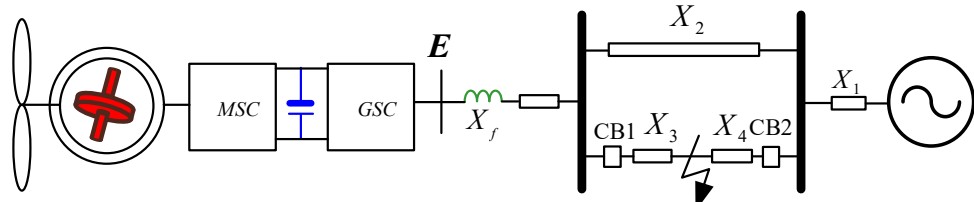

**Figure 6.** Single type-4 wind turbine infinite-bus (SMIB) system.

The transient analysis idea during fault recovery is shown in Figure 7. It is assumed that the system achieves a stable state at point a during grid fault. When the fault is cleared, the system goes through the transient process from point a to equilibrium point c after grid fault. However, due to the transient switch control introduced by active power climbing, the transient process is divided into two stages: during and after active power climbing. In the two stages, the network equations are the same. However, the motion equation models of direct drive wind turbines are different, resulting in state trajectories that are dominated by different dynamic equations. In the stage of active power climbing, the state trajectory moves from point a driven by the motion equation model and network equation in the stage of active power climbing. When the active current order reaches about the frozen value before grid fault, the stage ends, and the final state is the initial state of the second stage. After active power climbing, the system goes through the transition process from the final state in the stage of active power climbing to a stable equilibrium point after a grid fault. Due to the strong nonlinearity, the dynamic behavior and even stability issue in the second stage is significantly influenced by the final state in the stage of active power climbing. According to the attraction region theory of nonlinear system, there exists an attraction region in state space for the stable equilibrium point. Only if the initial state lies in the attraction region can the system keep transient stable. Otherwise, transient instability will occur. Since the initial state in the second stage during fault recovery is determined by

the final state in the first stage, the dynamic behavior in the stage of active power climbing will have much influence on the dynamic behavior and transient stability issue after active power climbing.

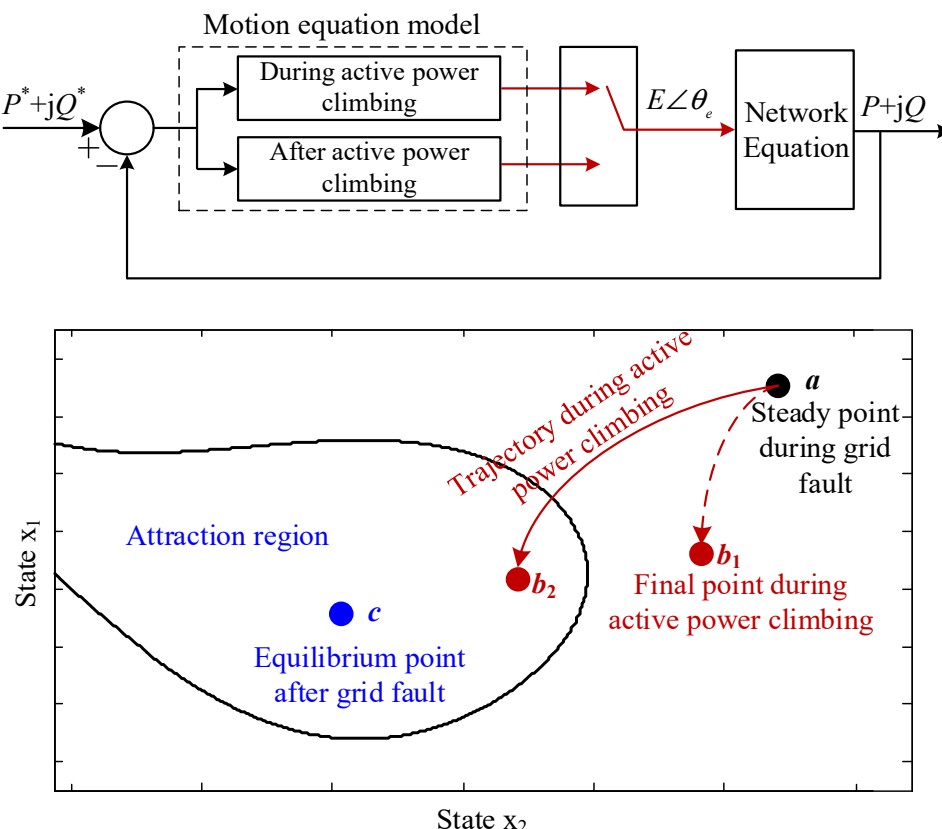

**Figure 7.** Transient analysis idea during fault recovery.

### 4.1. Transient Analysis in Stage of Active Power Climbing

In the stage of active power climbing, closed-loop dynamics of internal voltage can be investigated by combining the motion equation model and network model. Simplified network topology is shown in Figure 8, and the grid is represented by its Thevenin equivalent circuit. When considering transient behavior in the DC-link voltage control time scale, fast dynamics of the network are neglected, and an algebraic equation is used to calculate power through voltage vectors [8]. In Figure 9, it is shown that phase dynamics of internal voltage are induced by time-varying excitation from the active current order. In the model, the time-varying excitation can be further replaced by the integral calculus of constant $k_{ramp}$.

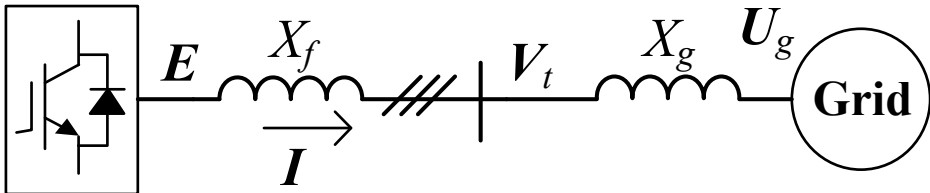

**Figure 8.** Simplified network topology.

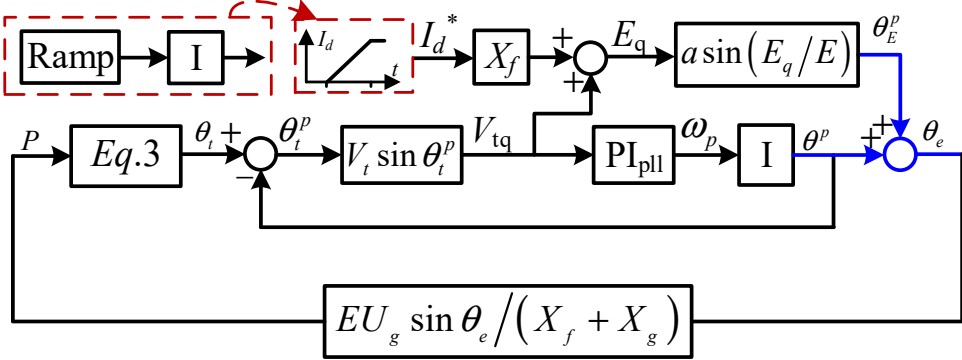

**Figure 9.** Phase dynamics of internal voltage in stage of active power climbing.

It is known that phase of internal voltage is composed of two parts: $\theta_p$ that is dominated by PLL and $\theta_E^p$ of internal voltage relative to PLL. Both of them are induced by active current order excitation. Since $\theta_p$ is directly related with $V_{tq}$, its dynamics can be investigated based on the relationship of $V_{tq}$ and the active current order. Utilizing terminal voltage $V_t$ and grid voltage $U_g$ in Figure 8, active power can be represented by

$$P = \frac{V_t U_g \sin \theta_t}{X_g} \tag{10}$$

In addition to this, active power can also be calculated through $d$-axis and $q$-axis current, which is obtained by

$$P = U_g I_d^p \cos \theta_p - U_g I_q^p \sin \theta_p \tag{11}$$

Since the phase of terminal voltage is also composed of two parts: $\theta_p$ and $\theta_t^p$ of terminal voltage relative to PLL, $V_t \sin \theta_t$ in (10) has another form represented by

$$V_t \sin \theta_t = V_t \sin \theta_t^p \cos \theta_p + V_t \cos \theta_t^p \sin \theta_p \tag{12}$$

Combing (10)–(12), relationship of $V_{tq}$ and active current order excitation is obtained by

$$V_{tq} = X_g \int k_{ramp} dt - U_g \sin \theta_p \tag{13}$$

Further, differentiating (13), the following expression can be obtained.

$$\frac{dV_{tq}}{dt} = k_{ramp} X_g - \omega_p U_g \cos \theta_p \tag{14}$$

Then the frequency dynamics of internal voltage dominated by PLL can be shown in Figure 10. It is a step response of a third-order nonlinear dynamical system, and excitation is related with $X_g$ and $k_{ramp}$. Nonlinearity exists in red dashed line frame in Figure 10. In addition to these, the initial states in Figure 10 reflect the influence of states during grid fault on the step response, and the initial states of $\theta_p$ and $V_{tq}$ are represented by (15) and (16), which is calculated based on the state-equation during grid fault [13]. Since it is assumed that steady states are achieved during grid fault, the initial integral state of PLL's PI controller is usually zero.

$$\theta_{p\_initial} = a \sin \left( \frac{X_{gf} I_{df0}}{U_{df}} \right) \tag{15}$$

$$V_{tq\_initial} = X_g I_{df0} - U_g \sin \left( \theta_{p\_initial} \right) \tag{16}$$

where $U_{df}$ and $X_{gf}$ are Thevenin equivalents of grid and $I_{df0}$ is the active current order, which are all during grid fault.

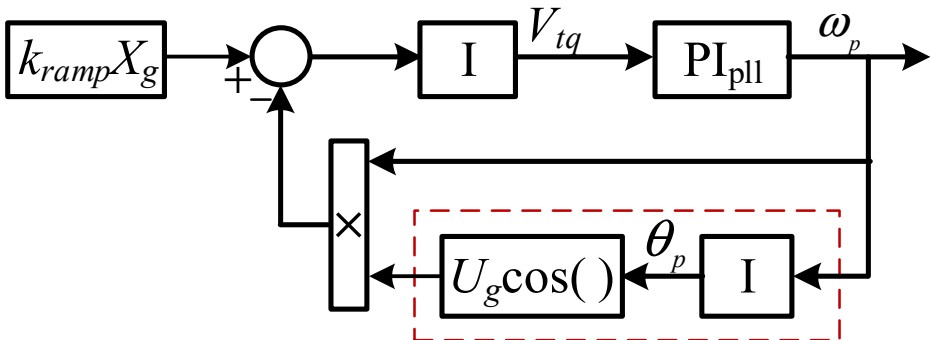

**Figure 10.** Frequency dynamics of internal voltage dominated by PLL.

For convenience, assume that deep voltage sag is considered and the active current order during grid fault is zero. Thus, $V_{tq}$ will not jump and keep zero at the beginning of fault recovery. Since the initial state of $\theta_p$ is also zero, $V_{tq}$ begins to increase driving by $k_{ramp}X_g - \omega_p U_g cos\theta_p$, and then $\omega_p$ and $\theta_p$ both increase from the zero initial state. However, in a short period of time, $\theta_p$ is very small and $U_g cos\theta_p$ is approximate to be constant $U_g$. Thus, the nonlinear part in the red dashed line frame can be replaced by a constant and $\omega_p$ is approximate to be the step response of the second-order system as represented by

$$\omega_{p\_approximation} = L^{-1}\left[\frac{k_{p\_pll}s + k_{i\_pll}}{s^2 + k_{p\_pll}s + k_{i\_pll}}\frac{k_{ramp}X_g}{s}\right] \tag{17}$$

As time prolongs, $\theta_p$ becomes large, and the influence of the nonlinear part should be considered. At this time, due to the fast response of PLL, dynamical regulation resulted in a large deviation of $k_{ramp}X_g$ and $\omega_p U_g cos\theta_p$ can be thought to be finished and approximation of $k_{ramp}X_g \approx \omega_p U_g cos\theta_p$ is reasonable. Thus, the dynamics of $\omega_p$ can be represented by the quasi-steady-state solution shown below.

$$\omega_{p\_quasi\_steady\_state} = \frac{k_{ramp}X_g}{U_g \cos\left(\int \omega_p\right)} \tag{18}$$

Based on the above, the dynamics of $\omega_p$, in the whole stage of active power climbing, can be approximately represented by

$$\omega_p \approx \omega_{p\_approximation} + \omega_{p\_quasi\_steady\_state} - k_{ramp}X_g \tag{19}$$

In the initial stage, it can be depicted by the step response of the second-order system, and then the quasi-steady-state solution reflects the subsequent dynamics. Further, the quasi-steady-state solution also has an approximate relationship represented by

$$\int (k_{ramp}X_g)dt \approx \int (\omega_p U_g \cos\theta_p)dt \tag{20}$$

Then $\theta_p$ at the end of active power climbing can be estimated by

$$\theta_{1s} = a\sin\left(\frac{X_g I_{d0}}{U_g}\right) \tag{21}$$

$\omega_p$ reaches the maximum at this time, which is represented by

$$\omega_{p\_max} \approx \frac{(k_{ramp}X_g)}{(U_g \cos\theta_{1s})} \tag{22}$$

Simulated results verified the analysis is shown in Figure 11. Since $\theta_p$ is integral of $\omega_p$, dynamics of $\theta_p$ is charactered by monotonous increase. Further analysis reveals that the influence of states of amplitude branch on the oscillation in the second stage is very small. Thus dynamics of the amplitude branch in the first stage will not be deeply investigated.

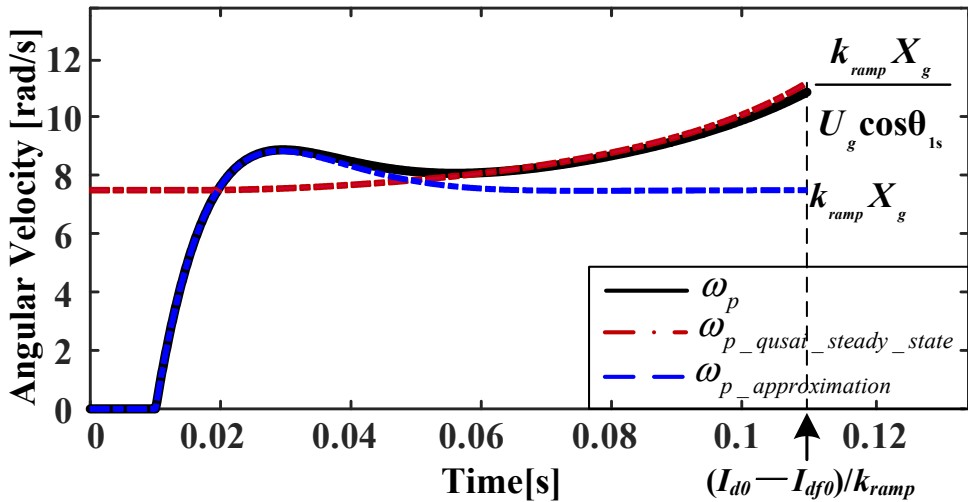

**Figure 11.** Frequency response of internal voltage dominated by PLL.

From Figure 10, active power climbing rate $k_{ramp}$ has much influence on frequency response of internal voltage dominated by PLL. By numerical calculation, the frequency response at different active power climbing rates is shown in Figure 12. It is seen that the frequency offset tends to be large at the end of active power climbing with the increase in active power climbing rate. Since the final state in the stage determines the initial state after active power climbing, it is indicated that the initial state in the second stage will deviate from the equilibrium point far away with the increase in active power climbing rate, which will deteriorate the transient behavior and even bring transient instability issue after active power climbing.

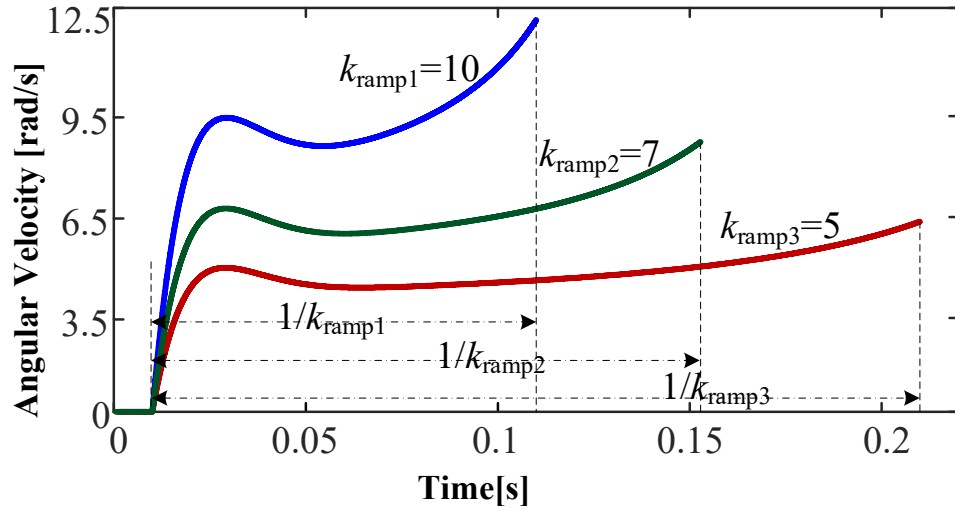

**Figure 12.** Influence of $k_{ramp}$ on frequency response of internal voltage.

### 4.2. Nonlinear Oscillation Analysis after Active Power Climbing

Based on the motion equation model in Figure 5, it is known that the open-loop characteristics of a wind turbine are depicted by two input and two output nonlinear transfer functions. In order to qualitatively investigate the influence of nonlinearity on large-signal oscillation behavior, single input and single output dynamical equation are employed for convenience based on a hypothesis. Here open-loop phase dynamics induced by unbalanced active power are investigated, as shown in Figure 13.

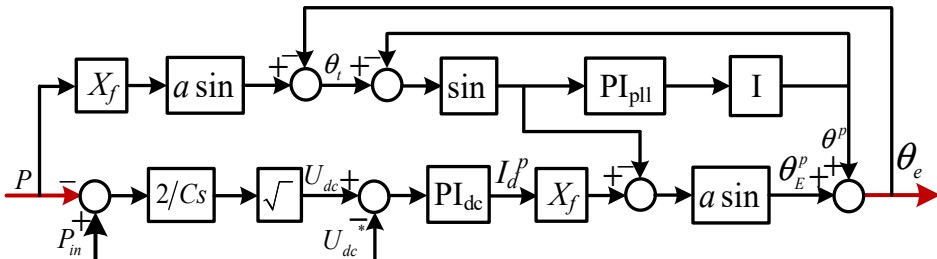

**Figure 13.** Open-loop phase dynamics with amplitude dynamics neglected.

The nonlinear function has three types: sin, asin, and square root. The influence of nonlinear function on open-loop characteristics can be reflected by (23). The small increase in the amount of output relative to the small increase in the amount of input at different operating points at a disturbed trajectory is different. The complexity induced by nonlinearity has resulted from this.

$$\sin(x_i + \Delta x) = \sin x_i + (\cos x_i)\Delta x \tag{23}$$

However, based on the geometric interpretation of the Euler integral, short time window dynamics around any operating point at a disturbed trajectory can be represented by a time-varying linear equation and constant excitation. The linear equation is obtained by linearization at the studied operating point, which usually is not the equilibrium point. Thus, open-loop characteristics in a short time window can be represented by (24).

$$\Delta\theta_e = G(s, X_i)\Delta P \tag{24}$$

Since $G(s, X_i)$ is related with operating point $X_i$, it is not constant and changes with time. When the disturbance is small, it means that $X_i$ is very close to the equilibrium point $X_e$ and the influence of change of $X_i$ on $G(s, X_i)$ is so small that it can be neglected. As a result, $G(s, X_i)$ is fixed, and small-signal dynamics have constant oscillation modes. Amplitude attenuation and the oscillation frequency are fixed. However, when the disturbance is large, $G(s, X_i)$ changes a lot with time. It is known that short time window oscillation characteristics are related with $G(s, X_i)$. Thus large-signal oscillation characteristics may be very different from small-signal oscillation, and its amplitude attenuation and oscillation frequency are not fixed.

$$\sin(x_e + x) = \sin x_e + (\cos x_e)x - 0.5(\sin x_e)x^2 + O\left(x^2\right) \tag{25}$$

Further, the influence of nonlinear function on open-loop characteristics can be investigated from the viewpoint of Taylor's high-order expansion. From (25), it is known that the output of a nonlinear function has other frequency components even if the input is a single frequency sinusoidal signal, and as the amplitude of the input signal increases, other frequency components in the output signal tend to be large. Due to these characteristics, closed-loop oscillation behavior will be more complex.

In order to study closed-loop oscillation behavior, the motion equation model in Figure 5 is combined with the network model in Figure 8. Oscillatory modes of linearized system at equilibrium point are listed in Table 1. It is shown that the linearized system

is poor damping and mode three dominates the small-signal oscillation. Comparative simulated results shown in Figure 14 reveals that large signal oscillation characteristics (amplitude attenuation and oscillation frequency) are different from that of linear oscillation. In order to further investigate the large-signal oscillation characteristics, time-frequency analysis based on the Hilbert transform [31] is employed.

**Table 1.** Oscillatory modes of equilibrium point after active power climbing.

| Mode | Eigenvalue | Freq.(Hz) | Damping Ratio |
|------|------------|-----------|---------------|
| 1 | $-100 \pm 99\mathrm{j}$ | 15.8 | 71% |
| 2 | $-29.4 \pm 35\mathrm{j}$ | 5.6 | 64% |
| 3 | $-0.6 \pm 67.9\mathrm{j}$ | 10.8 | 1.3% |

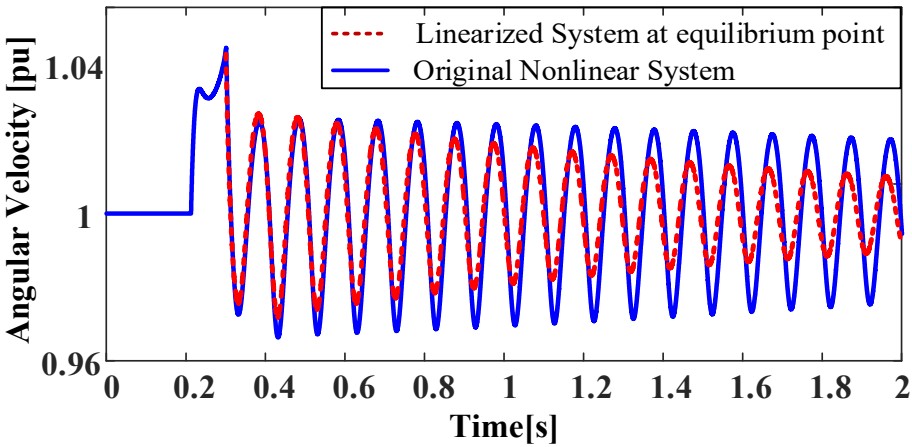

**Figure 14.** Comparative simulated results of internal voltage's angular velocity between original nonlinear system and linearized system at equilibrium point.

Since damping ratios of modes one and two are large, the oscillation component dominated by them will attenuate quickly, and the obtained simulated responses can be thought to be a single component from the viewpoint of a nonstationary signal. The single-component large-signal oscillation has the unified form represented by

$$x(t) = A e^{\int \alpha(t)dt} \cos\left(\int \omega(t)dt\right) \tag{26}$$

Here $\alpha(t)$ and $\omega(t)$ are defined as instantaneous amplitude attenuation ratio and oscillation frequency, respectively. If the oscillation is linear, $\alpha(t)$ and $\omega(t)$ are constantly determined by mode three. However, due to nonlinearity, $\alpha(t)$ and $\omega(t)$ change with time. In order to obtain $\alpha(t)$ and $\omega(t)$, Hilbert transform of $x(t)$ is utilized and $y(t)$ can be attained represented by

$$y(t) = A e^{\int \alpha(t)dt} \sin\left(\int \omega(t)dt\right) \tag{27}$$

Based on $x(t)$ and $y(t)$, the amplitude dynamics $A(t)$ and phase dynamics $\theta(t)$ can be represented by

$$A(t) = A e^{\int \alpha(t)dt} = \sqrt{x(t)^2 + y(t)^2} \tag{28}$$

$$\theta(t) = \int \omega(t)dt = \arctan\left(\frac{y(t)}{x(t)}\right) \tag{29}$$

Then $\alpha(t)$ and $\omega(t)$ can be calculated by

$$\alpha(t) = \frac{dA(t)/dt}{A(t)} \tag{30}$$

$$\omega(t) = \frac{d\theta(t)/dt}{\theta(t)} \tag{31}$$

Based on the Hilbert transform, the obtained instantaneous amplitude attenuation ratio $\alpha(t)$ and oscillation frequency $\omega(t)$ are shown in Figure 15. It is known that $\alpha(t)$ and $\omega(t)$ are both not constant and change with time. Further, $\alpha(t)$ and $\omega(t)$ oscillates around mode 3. This also verifies the idea of piecewise linearization with short time window. Since the short time window open-loop characteristics are determined by $G(s, X_i)$ and states $X_i$ oscillates around equilibrium point, instantaneous amplitude attenuation ratio, and oscillation frequency are inevitable to change around mode three with time. Figure 15 also shows that $\alpha(t)$ varies a lot around real part of mode three. This reveals that $\alpha(t)$ is very sensitive to change of states. Integral of $\alpha(t)$ reflects attenuation of amplitude. Figure 16 shows that $\int \alpha(t)dt$ tends to be larger than $\int \alpha_0 dt$ as time increases. This reveals that the nonlinearity deteriorates amplitude attenuation.

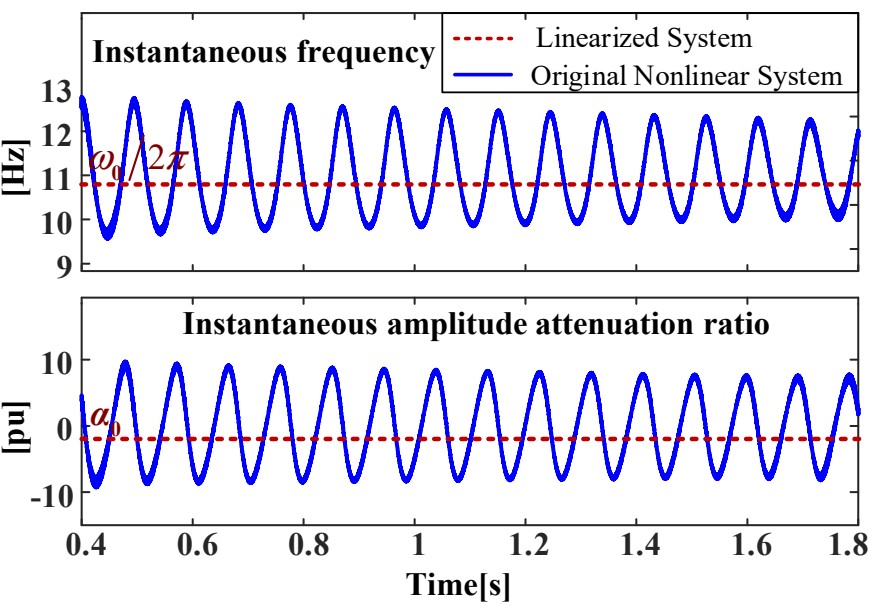

**Figure 15.** Instantaneous frequency and amplitude attenuation ratio characteristic.

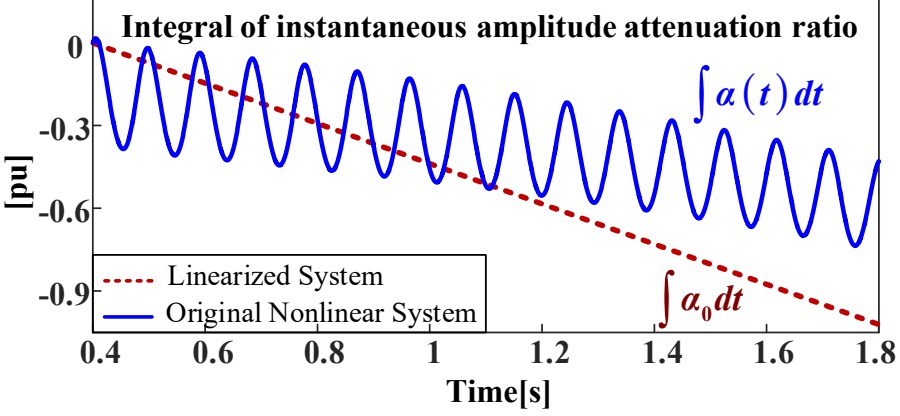

**Figure 16.** Integral of instantaneous amplitude attenuation ratio.

Time-frequency analysis digs deep into the nonlinear oscillation characteristics. In order to further carry out the mechanism explanation, Taylor's high-order expansion joint with the analysis idea of Normal Form is employed. From (25), it is known that Taylor's high-order expansion can achieve a good approximation of nonlinear function, and the order of high order term is related to the disturbance. Here second-order approximation is considered, and dynamics of the $j$th state can be represented by

$$\frac{dx_j}{dt} \approx f_j(X_e) + A_j(X - X_e) + 0.5(X - X_e)^T H^j(X - X_e) \tag{32}$$

Where $A_j$ is the $j$th row of the Jacobian matrix $[\partial f / \partial X]$, and $H^j$ is the Hessian matrix. Comparative simulated results among the original nonlinear system, first-order and second-order approximated systems are shown in Figure 17. It is known that second-order approximation almost achieves the same dynamical response as that of the original nonlinear system.

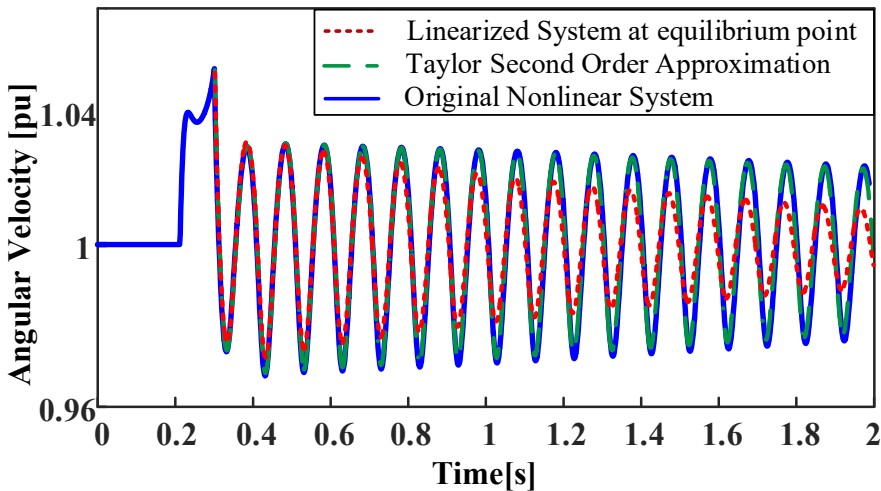

**Figure 17.** Comparative simulated results among original nonlinear system, first order and second order approximated system at equilibrium point.

Inspired by the analysis idea of Normal Form, analysis of the second-order approximated system can further explain nonlinear oscillation characteristics. The idea of Normal Form is that by the nonlinear transform of state variables, second-order terms in a state-space equation can be eliminated and an approximate solution of the nonlinear system is obtained, and the oscillation behavior is dominated by individual system modes, $\lambda_1, \lambda_2, \cdots, \lambda_n$ that are calculated by Jacobian matrix and second-order modes, $\lambda_1 + \lambda_1, \lambda_1 + \lambda_2, \cdots, \lambda_{n-1} + \lambda_n, \lambda_n + \lambda_n$. However, the base of the solution of Normal Form is still eigenvalues of the linearized system at the equilibrium point, and the obtaining of an approximate solution is under the condition that the influence of higher-order terms is neglected. These are reasons for the approximate solution's error. Since the approximate solution is not the target here and just an analysis idea is employed, the base of the solution can be selected in aid of Fourier and prony analysis, and oscillation behavior is still dominated by individual system modes and second-order modes. Fourier spectra in Figure 18 reveal that the second-order mode exists, and its frequency is almost twice the base dominant mode's. The second-order mode results from the nonlinear modal interaction of the base dominant mode. Due to the existence of the second-order mode component, the instantaneous oscillation frequency changes with time can be explained, which can also be understood from (33) and Figure 19. Further, prony analysis results in Table 2 show that the nonlinear oscillation is dominated by two modes: poor damping base mode and second-order mode, which is the combination of the poor damping base mode.

The real part of the base mode is smaller than that of the linear system. This further verifies that nonlinearity deteriorates amplitude attenuation.

$$
\begin{aligned}
x(t) = A e^{\int \alpha(t)dt} &\cos(\int \omega(t)dt) \\
&\approx A_1 e^{-\alpha_1 t}\cos(\omega_1 t + \theta_1) + A_2 e^{-2\alpha_1 t}\cos(2\omega_1 t + \theta_2)
\end{aligned}
\tag{33}
$$

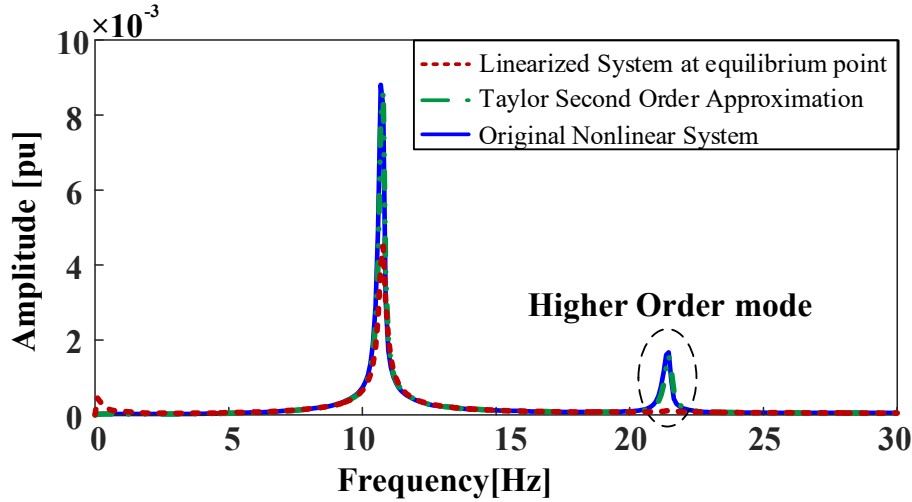

**Figure 18.** Fourier spectra of simulated results.

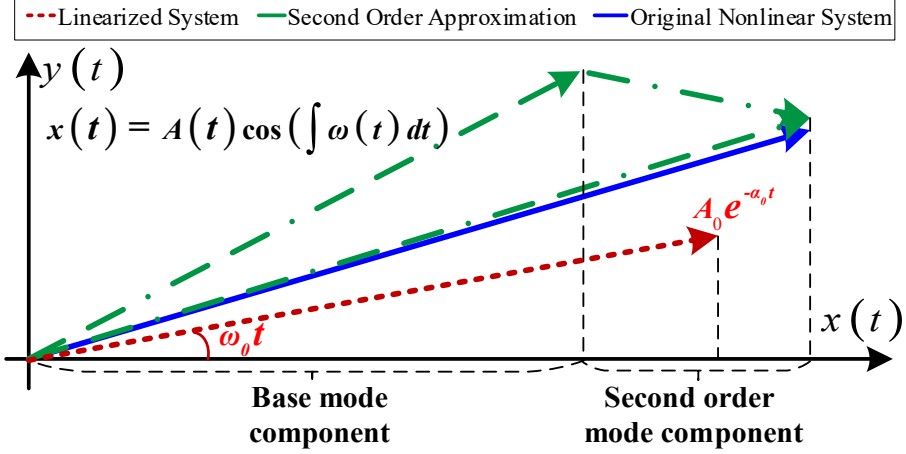

**Figure 19.** Mechanism explanation of instantaneous oscillation frequency changing with time.

**Table 2.** Prony analysis results of nonlinear oscillation response.

| Mode | Eigenvalue | Freq.(Hz) | Damping Ratio |
|------|-----------|-----------|---------------|
| 3 | $-0.32 \pm 60\mathrm{j}$ | 10 | 0.53% |
| 3,3 | $-0.75 \pm 127\mathrm{j}$ | 21 | 0.59% |

*4.3. Influence of Ramp Rate Limit in First Stage on Oscillation Behavior in Second Stage*

Based on the above analysis, it is known that the large signal oscillation behavior after active power climbing is very different from linear oscillation. Due to the influence of nonlinearity, its instantaneous amplitude attenuation ratio and oscillation frequency change with time, and the size of fluctuation is related to the state at the end of active power climbing. The above analysis further reveals that the comprehensive effect of the time-varying amplitude attenuation ratio is to deteriorate amplitude attenuation. When the initial state is far away from the equilibrium point in the second stage, oscillation with increasing amplitude may occur.

Since the initial state after active power climbing is dependent on the final state during active power climbing, transient behavior during the stage of active power climbing influences the subsequent oscillation. Based on the analysis in Section 4.1, it is known that initial states after active power climbing tend to be far away from the equilibrium point when $k_{ramp}$ increases. As a result, when $k_{ramp}$ is large, the influence of nonlinearity on large-signal oscillation behavior is strong. Since the comprehensive effect of nonlinearity is to deteriorate amplitude attenuation based on the analysis in Section 4.2, it is indicated that the nonlinear oscillation after active power climbing decays slowly and even diverges with the increase in active power climbing rate. Comparative simulated results at different ramp rate limit based on MATLAB$^{®}$ is shown in Figure 20. It is seen that the frequency offset at the end of active power climbing tends to be large, and then the subsequent oscillation after active power climbing tends to decay slowly and even diverges with the increase of active power climbing rate, which verifies the above theoretical analysis.

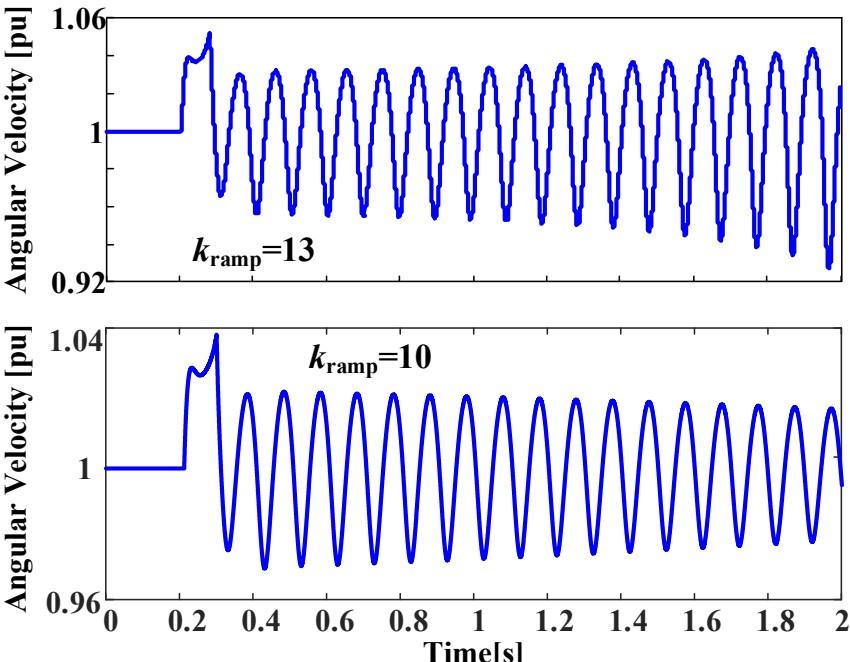

**Figure 20.** Comparative simulated results at different ramp rate limit.

## 5. Conclusions

In this paper, transient stability of direct drive wind turbine in DC-link voltage control timescale during LVRT is studied, with practice transient switch control considered. A novel two-stage motion equation model that depicts the phase and amplitude dynamics of internal voltage driven by unbalanced active and reactive power is developed firstly to physically study the transient characteristics of a direct drive wind turbine. Then the transient behavior during fault recovery is explored based on the developed model. When considering discontinuity resulting from the transient switch control, the whole transient process during fault recovery is divided into two stages: during and after active power climbing. In the first stage, frequency dynamics of a direct drive wind turbine at the excitation of active power climbing are studied. A novel approximate analytical expression is proposed to clearly reveal the transient frequency response and the influence of the active power climbing rate on it. After active power climbing, a novel analysis idea combining time-frequency analysis based on the Hilbert transform and high order modes is employed to investigate and reveal the nonlinear oscillation. The influence of transient behavior in the stage of active power climbing on the nonlinear oscillation after active power climbing is also explored. The conclusions and key findings are as follows.

(1) During active power climbing, an approximate monotonic increase in the wind turbine's angular frequency is identified at the excitation of active power climbing. With the increase in active power climbing rate, the frequency offset at the end of active power climbing tends to be large.

(2) After active power climbing, nonlinear oscillation characterized by time-varying oscillation frequency and amplitude attenuation ratio is revealed. It is found that the comprehensive effect of the time-varying amplitude attenuation ratio is to deteriorate amplitude attenuation. When the initial state tends to be far away from the equilibrium point in this stage, the nonlinear oscillation tends to decay slowly and even diverge, bringing in transient instability.

(3) The final state during active power climbing determines the initial state after active power climbing. With the increase in active power climbing rate, the final state during active power climbing will deviate from the equilibrium point after active power climbing far away. Then the amplitude attenuation of the nonlinear oscillation deteriorates, and oscillation with increasing amplitude is easier to occur.

**Author Contributions:** Conception and analysis, Q.H.; simulation and prony analysis, C.L.; high order modal analysis, G.W.; writing—original draft preparation, Q.H.; writing—review and editing, Y.X. and C.L.; literature research, Y.M. All authors have read and agreed to the published version of the manuscript.

**Funding:** This work was supported by National Natural Science Foundation of China under Grant 52107137.

**Institutional Review Board Statement:** Not applicable.

**Informed Consent Statement:** Not applicable.

**Data Availability Statement:** Not applicable.

**Conflicts of Interest:** The authors declare no conflict of interest.

## Nomenclature

| Symbol | Explanation |
| --- | --- |
| $E$ | GSC internal voltage vector |
| $V_t$ | Terminal voltage vector |
| $U_g$ | Equivalent grid voltage vector |
| $I$ | Current vector across filter inductor |
| $P,Q$ | Active and reactive power output of GSC |
| $X_g$ | Equivalent grid inductor |
| $X_f$ | Grid-side filter inductor |
| $\theta_p$ | PLL output angle relative to grid voltage |
| $\omega_p$ | Angular velocity of PLL relative to grid voltage |
| $k_{ramp}$ | Ramp rate limit |
| $T_{rv}, k_v$ | Parameters of AVC's controller |
| $k_{p\_dc}, k_{i\_dc}$ | Parameters of DVC's PI controller |
| $k_{p\_pll}, k_{i\_pll}$ | Parameters of PLL's PI controller |
| *Subscripts: dq* | components in PLL reference frame |

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
