# Peer review of "Transient Stability Analysis of Direct Drive Wind Turbine in DC-Link Voltage Control Timescale during Grid Fault"

_processes, doi:10.3390/pr10040774_

Round 1

Reviewer 1 Report

The authors are kindly asked to take into account the following comments:

Q1- Figure 1 should be revised. The role of the blocks of “PLL” and “Hysteresis control of chopper” is not clear.

Q2- The switching mechanism in Figure 2 should be presented.

Q3- Where are the initial condition values captured in Equations 15 and 16 used for?  What is their value for the simulated problem?

Q4- The type of software employed is not clear. The authors should use the symbol ® or TM for the software employed, for example MATLAB® (The MathWorks, Inc., Natick, MA, USA).

Q5- Before designing the control system, it is necessary to provide an appropriate analysis of dynamic behaviors, nonlinearity, stability conditions, non-minimal phase behaviors, and interactions between the system variables. Procedure of designing control system is unclear.

Q6- Useless information is presented in the conclusion section. Please highlight the contributions and key findings.

Reviewer 2 Report

The subject of the article is very specialized and is a sensitive and vital topic in the power system. Do the following to improve your work.

  1. Use numerical results in writing an abstract.
  2. The literature and structure of the article is similar to the following article, and it is better for the authors to refer to the following work and express the difference between their work.  J. Hu, Q. Hu, B. Wang, H. Tang and Y. Chi, "Small Signal Instability of PLL-Synchronized Type-4 Wind Turbines Connected to High-Impedance AC Grid During LVRT," in IEEE Transactions on Energy Conversion, vol. 31, no. 4, pp. 1676-1687, Dec. 2016, doi: 10.1109/TEC.2016.2577606.
  3. It is better to compare in simulation with at least one similar task. 
  4. You could use methods based on computational intelligence to make your work better and more accurate, or at least have an overview. https://doi.org/10.1016/j.asej.2021.08.007 , https://doi.org/10.1177/01423312211048038 or etc.
  5. The references are mostly old and it is better to use more references from the last three years. 

Reviewer 3 Report

The paper present an interesting work, well presented by authors. The introduction/background is well presented, reviewing several relevant and similar studies. But the references should be updated, only 3 titles are newer than 5 years old and none is less than 3 years old.

The authors use “type-IV wind turbines” and “Type-4 Wind Turbine”, it is better to use only one form in the whole paper. Also in Figure 3 there is used small s and capital S – are they the same notation?

All abbreviations used in the paper should be explained, for example “ LVRT”.

Round 2

Reviewer 2 Report

All comments have been answered correctly.

Author Response

Thank you for your  recognition of our research work.